# Micro- and Nanostructures of Agave Fructans to Stabilize Compounds of High Biological Value via Electrohydrodynamic Processing

**DOI:** 10.3390/nano9121659

**Published:** 2019-11-21

**Authors:** Carla N. Cruz-Salas, Cristina Prieto, Montserrat Calderón-Santoyo, José M. Lagarón, Juan A. Ragazzo-Sánchez

**Affiliations:** 1Laboratorio Integral de Investigación en Alimentos, Tecnológico Nacional de México/Instituto Tecnológico de Tepic, Av. Tecnológico 2595, Tepic C.P. 63175, Nayarit, Mexico; ccruz@ittepic.edu.mx (C.N.C.-S.); montserratcalder@gmail.com (M.C.-S.); 2Novel Materials and Nanotechnology Group, IATA-CSIC, Calle Catedrático Agustín Escardino Benlloch 7, 46980 Paterna, Spain; cprieto@iata.csic.es (C.P.); lagaron@iata.csic.es (J.M.L.)

**Keywords:** HDPAF, electrospinning, micro-nanofibers, β-carotene, thermoprotection, photoprotection

## Abstract

This study focuses on the use of high degree of polymerization agave fructans (HDPAF) as a polymer matrix to encapsulate compounds of high biological value within micro- and nanostructures by electrohydrodynamic processing. In this work, β-carotene was selected as a model compound, due to its high sensitivity to temperature, light and oxygen. Ultrafine fibers from HDPAF were obtained via this technology. These fibers showed an increase in fiber diameter when containing β-carotene, an encapsulation efficiency (EE) of 95% and a loading efficiency (LE) of 85%. The thermogravimetric analysis (TGA) showed a 90 °C shift in the β-carotene decomposition temperature with respect to its independent analysis, evidencing the HDPAF thermoprotective effect. Concerning the HDPAF photoprotector effect, only 21% of encapsulated β-carotene was lost after 48 h, while non-encapsulated β-carotene oxidized completely after 24 h. Consequently, fructans could be a feasible alternative to replace synthetic polymers in the encapsulation of compounds of high biological value.

## 1. Introduction

Obtaining micro- and nanostructures in the food industry represents a viable option for the incorporation and stabilization of compounds of high biological value (CHBV). The production of these structures containing CHBV is based on the encapsulation processes in which the bioactive is surrounded by polymeric materials that act as matrices and help to preserve their properties. Currently, there is a huge interest in the use of biopolymers that replace synthetic polymers in the encapsulation and transport processes of compounds of high biological value, as well as expanding their use in biomedical and pharmaceutical technologies [1,2]. Fructans are a new class of heterodispersed biopolymers that are found in various plants, such as the *Agave tequilana* from the Agavaceae family, and serve as an important source of reserve carbohydrates in the plant. Five types of fructans have been identified in nature so far, which are classified into inulins, neo-inulins, levans, neo-levans and mixed fructans, depending on the type of bond [3,4]. Agave fructans are characterized by the presence of fructose units with a terminal glucose connected by bonds β (2-1) and β (2-6) and can present different degrees of polymerization, which are determined by the species [5,6,7] as well as the environmental conditions in which the agave is produced, stored and processed [8,9].

Toriz et al. (2007) [10] propose a chemical structure for the fructans of *Agave tequilana* Weber var. Azul, based on the combination of permethylation and reductive rupture techniques for identification. They mainly propose the distribution of two monomers, the β-D fructofuranose terminal type (22%), 1-linked (30.8%), 6-linked (21.2%), 1-6 linked (19.8%) and the α-D terminal glucopiranose (7.3%).

The use of high degree of polymerization agave fructans (HDPAF) has been implemented in recent years as a wall material in encapsulation processes such as spray drying, obtaining remarkable results. Ortiz-Basurto et al. (2017) [11] microencapsulated pitanga juice (*Eugenia uniflora* L.) by spray drying with HDPAF and maltodextrin as encapsulants. These authors reported similar behaviors between both materials. Alternatively, Farías Cervantes et al. (2016) [12] used 50% agave fructans, as an encapsulant to obtain blackberry powder by spray drying, mentioning that agave fructans, in addition to increasing the encapsulation efficiency, could add prebiotic properties and improve the physico-chemical characteristics of the powder.

However, conventional encapsulation processes, such as spray drying, require the use of high temperatures or toxic reactives that could affect the CHBV or compromise its application in a food or pharmaceutical product. The electrohydrodynamic processing, electrospinning or electrospraying, has demonstrated to be a promising alternative to encapsulate and stabilize compounds of high biological value, since it does not require severe conditions of temperature, pressure or aggressive chemicals [13]. This technology allows obtaining structures called fibers or particles at the micron, submicron or nano scale [14]. This technique uses a potential difference for the electro-stretching of a drop of polymer solution from a charged electrode (capillary or free surface) to a collector. Once the drop has gained enough electrical charge to overcome the surface tension and viscosity of the polymer, the drop is stretched, causing ultra-fine fibers to emerge from the polymer drop, forming the so-called Taylor cone [15]. The extent of each phase, whether direct or in drag motion, depends on the operating parameters as well as the physical properties of the polymer, such as surface tension, conductivity and viscosity [16]. The difference between electrospinning and electrospraying (electrohydrodynamic spraying) is based on the degree of molecular cohesion, which can be easily controlled by variation in the concentration of the polymer solution [17]. Ramos-Hernández et al. (2018) [18] managed to obtain spherical structures with a size distribution of 440 to 880 nm, as well as the thermostability and photostability of β-carotene encapsulated by electrospraying, using solutions with concentrations from 10% to 50% of HDPAF.

Electro-spun nanofibers have also been used in CHBV encapsulation processes, such as the encapsulation of bioactive compounds in zein fibers with application in food and pharmaceutical products [19]; however, up to our knowledge, the production of electro-spun micro-nanofibers of HDPAF has not been reported yet. This material could be a feasible alternative to replace synthetic polymers in the encapsulation of compounds of high biological value, as well as expanding their use in biomedical and pharmaceutical products.

Antioxidants are a good example of CHBV due to the high number of health benefits attributed to them and their high sensitivity to physico-chemical factors [20]. Carotenoids are an important member of the antioxidant family, being natural organic pigments present in plants and some photosynthetic organisms. Their consumption is associated with health benefits by reducing the incidence of cancer and heart disease, as well as improving ocular health [21]. β-carotene is one of the most common carotenoids in the functionality of food, supplements and pharmaceuticals due to its high, pro-vitamin A activity and antioxidant capacity. However, its use is sometimes limited due to its sensitivity to oxidation, especially when exposed to high temperatures, light, oxygen, acidic conditions, etc. [19,22].

The aim of this study was to evaluate the feasibility of agave fructans to obtain micro-nanofibers through the electrospinning process, using β-carotene as a model compound, with properties and characteristics, as a first approach, and to provide stability to compounds of high biological value for further applications in the food and pharmaceutical areas.

## 2. Materials and Methods

### 2.1. Materials

High degree of polymerization agave fructans (HDPAF) were obtained in the Laboratorio Integral de Investigación en Alimentos (LIIA) of Tecnológico Nacional de México/Instituto Tecnológico de Tepic, Nayarit, México, from native fructans provided by the company Campos Azules Co. (Mexico City, Mexico); β-carotene (>97.0% UV, C40H56) (Sigma Aldrich, Steinheim, Germany), 96% ethyl alcohol (CTR Scientific, Monterrey N.L, Mexico), Chloroform (trichloromethane) HPLC grade (Fermon Episolv, Monterrey N.L, Mexico), Tego^®^ SML (Evonik Industries AG, Essen, Germany).

### 2.2. Preparation and Characterization of Polymer Solutions

Polymeric solutions were prepared with 70% (*w*/*w*) of HDPAF in hydroalcoholic solution (ethanol-water, at 10% *w*/*w*) and 1% (*w*/*w*) of Tego^®^ SML as surfactant was added. The solution was homogenized under magnetic stirring for 45 min at room temperature. In the alcoholic fraction of the solution, β-carotene was added (1% *w*/*w*), protecting the mixture from light during the homogenization period with dark paper.

The characterization of the solutions consisted in determining the electrical conductivity, viscosity and surface tension. The electrical conductivity was analyzed with a multiparameter potentiometer Hanna Instruments HI-4521 (Melrose, MA, USA). The viscosity (η) was determined with a Discovery HR-1 Hybrid rheometer (TA Instruments, New Castle, DE, USA), equipped with geometry Smart Swap ™ with automatic detection. The cone and plate geometry option was selected (2°, 60 mm of diameter, 64 µm of gap) and the Peltier system for temperature stabilization was used. The surface tension was measured with the equipment Force tensiometer model K20 EasyDyne (KRÜSS GmbH, Hamburg, Germany), with the Wilhelmy Plate method.

### 2.3. Obtaining Fibers by Electrospinning

The electrospinning process was performed in a machine LE-10 brand Fluidnatek^®^ from BIOINICIA company (Valencia, Spain), which has a voltage power supply of 19 kV. The injector is based on a syringe pump with a flow from 200 µL/h, and has a distance from the injector to the collector of 20 cm, the collector is cylindrical stainless steel and it has a variable rotation speed from 500 rpm. The process parameters such as voltage, flow, distance and rotation speed of the collector were adjusted in preliminary tests according to the characterization and stability of the solutions.

### 2.4. Morphology Analysis through SEM

The morphology and size of the fibers obtained were both determined with the scanning electron microscopy (SEM) technique with a Hitachi-S-4800 device (Hitachi High-Technologies Corporation, Tokyo, Japan). Approximately 1.5 mg of sample was fixed with double-sided tape on the sample holder, coated with gold-palladium for 2 min, and an acceleration voltage of 10 kV was used. The determination of the size distribution based on the diameters of the structures was made with the SEM system software (Hitachi High-Technologies Corporation, Tokyo, Japan) with at least 100 measurements per sample.

### 2.5. Loading and Encapsulation Efficiency

The loading efficiency (LE) of β-carotene in the fibers was determined considering the amount of total extract and final extract by weight difference, obtained with thermogravimetry analysis (TGA) using the TRIOS software (TA Instruments, New Castle, DE, USA) and applying Equation (1). Where the total extract corresponds to the amount of extract added to the fibers, the final extract represents the amount of extract determined by TGA. Regarding the encapsulation efficiency (EE), initially the amount of extract loaded was determined with the application of Equation (2).

Subsequently, the amount of surface was calculated, performing a superficial washing of the fibers with a solvent related to β-carotene and not to the polymer, in order to determine the content of compounds of high biological value (CHBV) on the fiber surface. To this end, 1 mg of fibers was taken and suspended in 1 mL of trichloromethane, centrifuging at 10,000 rpm for 1 min, analyzing the supernatant obtained absorbance at 466 nm in a Varian brand Cary^®^ 50 UV-Vis spectrophotometer (Sydney, Australia). The β-carotene content is calculated according to the calibration curve performed, y = 3.6298x + 0.0059 (R^2^ = 0.9998). Finally, knowing the values of loading and surface extract, Equation (3) was applied to determine the encapsulation efficiency (EE).

Loading efficiency (LE)
(1)$$\%LE=\left(1-\left(\frac{total\;extract-final\;extract}{total\;extract}\right)\right)\times 100$$

Loading extract
(2)$$Loading\;extract=\frac{\%EC*total\;extract}{100}$$

Encapsulation efficiency (EE)
(3)$$\%EE=\frac{loading\;extract-surface\;extract}{loading}\times 100$$

### 2.6. Thermogravimetric Analysis (TGA)

The thermogravimetric analysis to determine the decomposition temperature of each component of the fiber, as well as to determine the LE and to demonstrate the thermoprotective effect of the materials on β-carotene, was performed on a TGA 550 (TA Instruments, New Castle, DE, USA) in a nitrogen atmosphere (N_2_), with a heating ramp of 25 to 600 °C at a speed of 5 °C/min. The results were analyzed with the TRIOS software (TA Instruments, New Castle, DE, USA).

### 2.7. UV Photostability

In order to analyze the oxidation kinetics of β-carotene, the fibers obtained were exposed to a simulator of sunlight. An Osram Ultra-vitalux lamp (300 W) (OSRAM, Munich, Germany) was utilized, which generates a mixture of radiation, using a quartz discharge tube and a tungsten filament [19]. It was placed in a sample holder 1 mg of standard β-carotene, and samples of the fibers with and without β-carotene, which were exposed to ultraviolet (UV) radiation for 48 h at 37 °C. Samples were taken every 6 h. A central segment of the exposed material was cut and dissolved in distilled water at a ratio of 1 mg/mL with magnetic stirring for 1 min. Subsequently, 1 mL of chloroform was added and centrifuged (10,000 rpm, 1 min). The absorbance of the organic phase at 466 nm was measured with the Varian brand Cary^®^ 50 UV-Vis spectrophotometer (Sydney, Australia). Chloroform was used as the target. The results, obtained in relation to oxidation, are reported based on the relative content of β-carotene (% absorbance).

### 2.8. Statistic Analysis

Data analysis was performed using the least significant digit (LSD) test for comparison of means with STATISTICA version 10 (StatSoft, Inc. (2011), (Tulsa, OK, USA)). All tests were performed in triplicate.

## 3. Results and Discussion

### 3.1. Physicochemical Characterization of Polymer Solutions

First, the physicochemical properties of the polymer solutions were evaluated, since the stability of the electrohydrodynamic process and the morphology obtained are highly related to them. The solutions presented viscosity values of 3.69 ± 0.05 and 3.36 ± 0.03 Pa·s without and with β-carotene, respectively, as shown is Table 1. These results are in the same order of magnitude as those obtained by Kutzli et al. (2019) [23], at high concentrations of maltodextrin combined with whey protein isolated (WPI) or soy protein isolated (SPI). They obtained values of apparent viscosity of 4.85 ± 0.14 Pa·s (WPI 80:5), 5.88 ± 0.18 Pa·s (WPI 80:10), 5.14 ± 0.09 Pa·s (SPI 80:5) and 7.77 ± 0.12 Pa·s (SPI 80:10), respectively. The similarity of fructans with maltodextrin with respect to this property is mainly due to their structural conformation. Concerning the surface tension, similar values in the solutions with and without β-carotene were obtained (29.6 ± 0.2 and 30.1 ± 0.1 mN/m), showing that the β-carotene incorporation does not modify the penetration resistance of the solution (*p* ≤ 0.05). Ramos-Hernández et al. (2018) [18] analyzed the surface tension of HDPAF solutions at different concentrations, obtaining a value of 23.46 mN/m for a 50% solution of the polymer, which differs from those obtained in this study. The difference could be associated with the type of surfactant used, but mainly at the concentration of the biopolymer (70%). In contrast, the addition of β-carotene to the polymer solution caused a significant increase in electrical conductivity from 5.54 ± 0.01 to 7.30 ± 0.03 mS/cm, and this can be attributed to the functional group loads of β-carotene (Table 1).

In the electrospinning process, the high concentrations of HDPAF allowed the increase of intermolecular interactions of the polymer with the solvent, and in the same way, the polymer–polymer crosslinking, favoring the stability of the flow [17]. In addition, polymer chain interactions have a close relationship with concentration and viscosity [24]. However, a high viscosity can cause clogging of the injector partially or totally, but it can also affect the morphology of the fibers due to the presence of artifacts in the structures.

### 3.2. Micro-Nanofiber Formation Process

The stabilization of the electrospinning process was achieved with a voltage of 19 kV, 200 µL/h feed flow of the polymer solution, 20 cm distance from the injector to the collector with 500 rpm rotation. A concentration of the polymer of 70% (*w*/*w*) was used, which allowed the formation of a network of polymer chains and enough entanglement to electro-stretch the solution, obtaining continuous fibers (Figure 1).

In preliminary studies, the electrospinning process was carried out with a 60% solution (*w*/*w*) of HDPAF concentration. The presence of structures, with artifacts that cannot be considered as spheres or fibers, was observed (Figure 1a). On the contrary, for a concentration of 70% of HDPAF, it was possible to obtain a homogeneous film on the collector. This is directly related to the increase in concentration, the structure of high-polymerization fructans and the molecular arrangement of the terminal chains and interactions that occur during the electrospinning process.

Despite not finding references in literature about fiber formation with HDPAF, some authors such as Lee et al. (2009) [25]; Kai et al. (2015) [26] reported the use of polysaccharides such as alginate, cellulose, chitosan, starch in the formation of fibers by electrospinning, which could be used as natural encapsulants in the area of medicine. HDPAFs have just been used so far for the production of spherical nanocapsules ([18]), which were obtained using a HDPAF concentration of 30% through the electrospraying process to encapsulate β-carotene. The possibility of obtaining fibers could increase the application field of agave fructans to medicine or biomaterials.

### 3.3. Morphology Analysis

Fibers obtained with solutions without β-carotene presented a smooth and continuous surface (Figure 1d) with a larger distribution of diameters, in the range of 750 to 1000 nm (Figure 1f). Structures obtained with β-carotene showed smooth, continuous surface fibers, with a slight fragmentation of some structures. This is possibly associated with the applied voltage and its effect on the breaking of the cross-links of the polymer chains. An increase in the diameter size distribution was observed, being in the range between 1250 and 1500 nm. The presence of pores and widenings in some segments was observed, which could be due to the lipophilic nature of β-carotene that limits cross-linking with the polymer or the relative humidity and vapor pressure of the solvent used. The high hygroscopicity of the HDPAF could also affect the diameter of the fibers. Bak et al. (2016) [27] analyzed the effect of relative humidity (30% and 60%) in the manufacture of collagen nanofibers. These authors reported that the quantity of humidity affects the morphological characteristics of the fibers obtained, and therefore concluded that at lower humidity the diameter of the fibers decreases.

### 3.4. Loading and Encapsulation Efficiency (LE and EE)

The encapsulation efficiency of the β-carotene inside the ultrathin fibers of HDPAF was 95%. This indicates that almost all the loaded compound was encapsulated in the center of the fibers. This value differs with the encapsulation efficiencies obtained with other polysaccharides, such as chitosan, where an encapsulation efficiency of β-carotene of 36% was obtained in the encapsulation by nanomicelles. However, it is similar to the encapsulation efficiency reported for the encapsulation of anthocyanins in xanthan gum in combination with starch (96%) by spray drying [28,29]. HDPAFs show similarities in their entrapment capacity compared to some proteins. In this sense, López-Rubio and Lagaron (2012) [30] report a 90% of encapsulation efficiency for the encapsulation of β-carotene in whey protein. Gomez-Estaca et al. (2012) [31] obtain an encapsulation efficiency between 85 and 90% in encapsules of curcumin into zein by electrospraying. The branched structure of the HDPAF favors the entrapment of the bioactive compound due to its available functional groups, forming bonds such as hydrogen bonds that stabilize the nucleus and maintain the compound of interest in the fiber. The loading efficiency was of 85% in the fibers based on thermogravimetric analysis.

### 3.5. Thermogravimetric Analysis

A thermogravimetric analysis was performed to the fibers as well as to each of the materials used to make the fibers, in order to study the thermal stability of the β-carotene inside the polymeric wall of HDPAF. The β-carotene displayed an initial decomposition temperature of 138 °C and a final temperature of 367 °C (Figure 2b), similar to the temperatures reported by Peinado et al. (2016) [32] when analyzing this powder compound with a maximum peak of decomposition at approximately 120 °C. In addition, there were two important variations in mass, which are related to the breakdown of the basic structure due to thermal energy, generating the formation of volatile organic compounds including 2-methyl-2-hepten-6-yne, 2-methyl-2-hepten-4-yne and β cyclocitral.

The thermal stability of the fibers in the absence of β-carotene (Figure 2b) was characterized by presenting a first weight loss of 5.0% at 100 °C, which is associated with the humidity content. The second variation occurred between 167 to 307 °C, attributed to HDPAF (57.91%), reaching the maximum decomposition temperature at 209 °C. This temperature is similar to that presented by the individually analyzed HDPAFs, which explains that the surfactant used does not modify the thermal properties of the polymer. Espinosa-Andrews and Urias-Silvas (2012) [33] reported decomposition temperatures of 200 to 222 °C for commercial agave fructans, with a similar thermal behavior to that reported in this study. Regarding the fibers with β-carotene, the thermogram showed three mass variations (Figure 2b).

The first corresponds to humidity (4.0%), the second, between 174 to 232 °C, is attributed to the breakdown of HDPAF according to Figure 2a, the third was in the temperature range of 232 to 288 °C, which was related with the breakdown of β-carotene, a higher degree of polymerization fructans and surfactant according to Figure 2b. This can be explained due to the nature of the compounds and their affinity between β-carotene and HDPAF. Likewise, the fibers with β-carotene showed a displacement of 90 °C in the initial decomposition temperature of this compound, demonstrating the thermoprotective effect of HDPAF.

### 3.6. Photostability Analysis

The need to evaluate the protection capacity of polymers used in encapsulation processes arises due to the susceptibility of photosensitive compounds to structural modification, such as β-carotene. The photostability of the fibers was evaluated under exposure of the fibers to ultraviolet A (UVA) light. According to the results, the photooxidation of uncapsulated β-carotene was presented from the beginning of the exposure, becoming total at 24 h.

On the contrary, when β-carotene was encapsulated within the fibers, the loss of its relative content was only 21% until 48 h (Figure 3). This behavior shows the ability of HDPAFs to reduce the photooxidation process, thus, prolonging the stability of β-carotene. Ramos-Hernández et al. (2018) [18] prepared capsules with HDPAF using a polymer concentration of 30% and 0.1% of β-carotene, and the results in the oxidation of β-carotene encapsulated with the electrospraying method showed a loss of 10% in capsules within 48 h. If the protective effect of both morphologies is compared, it is possible to observe that the fibers show twice the oxidation than the capsules, although the β-carotene content is 10 times higher in the fibers. The loss of photostability could be attributed to a reduced biopolymer-bioactive ratio, as well as to an effect of morphology, since the fibers have a larger surface exposed to light. For this reason, taking into account the amount of encapsulated β-carotene and the encapsulation efficiency obtained, the photostabilizing effect of the fiber could be considered very adequate.

In comparison with synthetic polymers, the behavior of the fibers with β-carotene is similar to that presented with the use of HDPAF. Peinado et al. (2016) [32] report a 20% loss of the relative content of the bioactive compound when exposed to UVA fibers made with polyethylene oxide (PEO) attributed to the stability of β-carotene once encapsulated, reflecting the polymer ability to limit oxygen diffusion and reduce exposure to light. On the other hand, de Freitas Zômpero et al. (2015) [34] obtain a 20% reduction after 6 h of exposure to UV light in fibers made with polyvinyl alcohol (PVOH) loaded with nanoliposomes, with the aim of analyzing the stability of the β-carotene. Whereas, the fibers in the present study show a reduction of 7% in the same period of time, which proves that HDPAFs have the ability to protect photosensitive compounds.

## 4. Conclusions

To the best of our knowledge, this paper reports for the very first time the possibility of generating continuous micro- and nanometric scale fibers from agave fructans of high degree of polymerization in 70% (*w*/*w*) solutions. This document also reports the possibility to encapsulate a bioactive model compound, such as β-carotene inside the fibers, obtaining a high encapsulation efficiency as well as providing stability to both temperature and UVA radiation.

The good results provided by fructans could suppose the expansion of its use and exploitations, since they could be a feasible alternative to replace synthetic polymers in the encapsulation of compounds of high biological value, as well as expanding their use in biomedical and pharmaceutical products. In this spirit, it is of important interest to consider the study of possible complexes with other polymers that help enhance their characteristics and properties, such as its high hygroscopicity, and structures more homogeneous, as well as their incursion into new applications.

## Figures and Tables

**Figure 1 nanomaterials-09-01659-f001:**
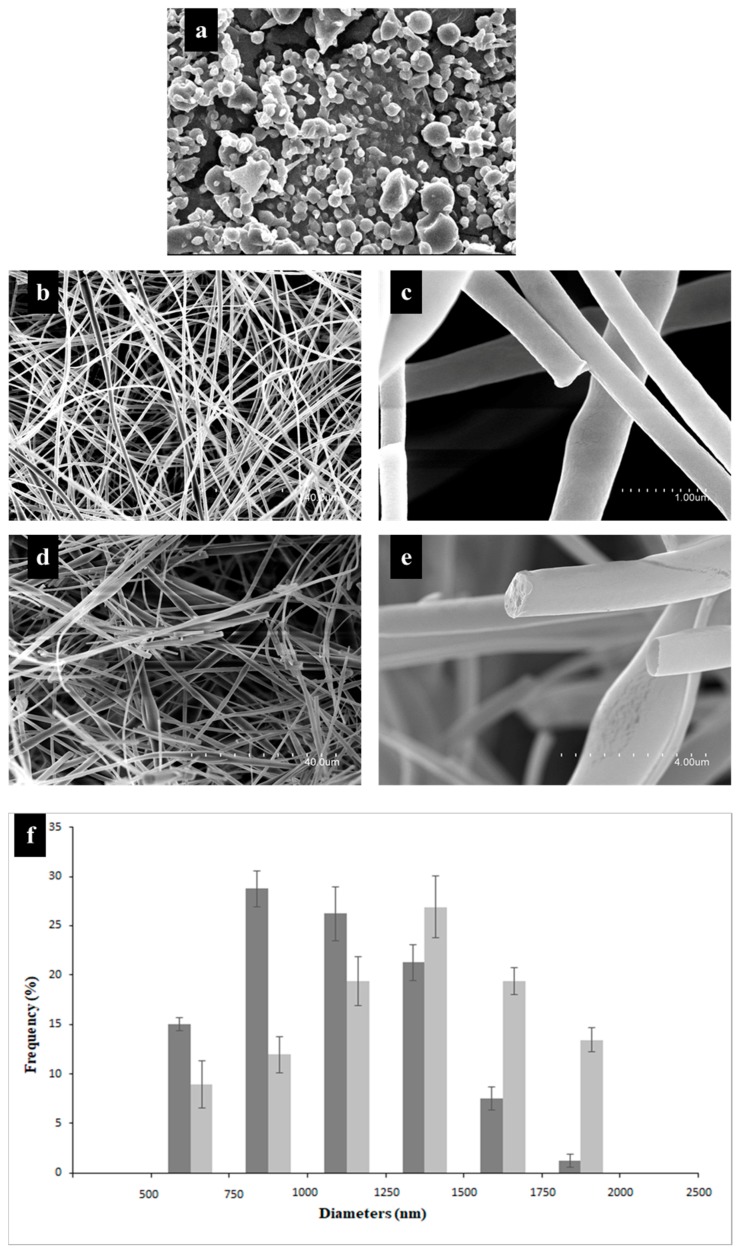
Scanning electron microscopy (SEM) micrographs show structures obtained with (**a**) 60% HDPAF, (**b**,**c**) fibers obtained with 70% HDPAF without β-carotene at different magnification, (**d**,**e**) fibers obtained with 70% HDPAF loaded with β-carotene at different magnification, (**f**) distribution of micro-nanofiber diameters obtained with the 70% HDPAF formulation (^▄^) without and (^▄^) with β-carotene. The average values and standard deviation (SD) were obtained from the analysis of three replicas.

**Figure 2 nanomaterials-09-01659-f002:**
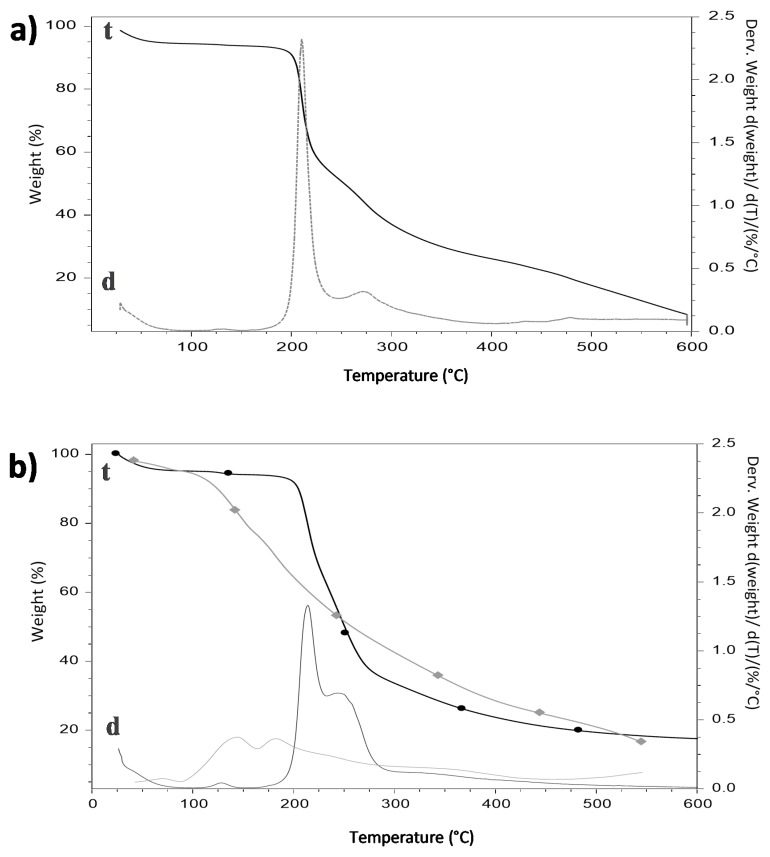
Thermograms of (**a**) HDPAF and (**b**) comparative (♦) β-carotene (●) fibers with β-carotene. (t, indicates thermograms and d, the derivatives of the thermograms).

**Figure 3 nanomaterials-09-01659-f003:**
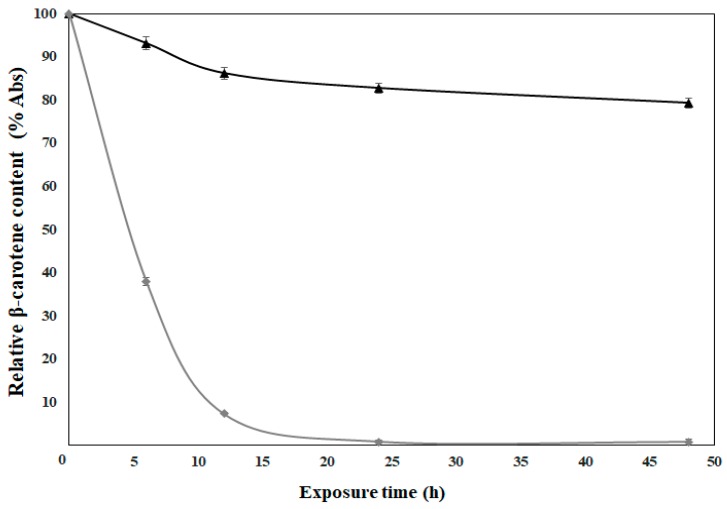
Photostability kinetics of the β-carotene (♦) and 70% HDPAF micro-nanofibers loaded with β-carotene (▲).

**Table 1 nanomaterials-09-01659-t001:** Physicochemical characterization of the 70% high degree of polymerization agave fructans (HDPAF) polymer solution with and without β-carotene.

Parameter	70% HDPAF without β-Carotene	70% HDPAF with β-Carotene
Viscosity ^1^ (Pa·s)	3.69 ± 0.05 ^a^	3.36 ± 0.03 ^a^
Surface tension (mN/m)	30.1 ± 0.1 ^b^	29.6 ± 0.2 ^b^
Electrical conductivity (mS/cm)	5.54 ± 0.01 ^c^	7.30 ± 0.03 ^d^

Different letters within the same row indicate significant differences among samples (α = 0.05). The average values were obtained from the analysis of three replicas. ^1^ Viscosity values were read at a shear rate of 39.8 s^−1^.

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
