# Peer review of "Micro- and Nanostructures of Agave Fructans to Stabilize Compounds of High Biological Value via Electrohydrodynamic Processing"

_nanomaterials, 2019, doi:10.3390/nano9121659_

Round 1

Reviewer 1 Report

This article deals with the preparation of b-carotene encapsulated a high degree of polymerization agave fructans (HDPAF) microfibers by electrospinning. Although the concept of the paper, i.e., the stabilization of the b-carotene by embedding in microfibers might be interesting, the experimental results are very poorly presented. More importantly, a similar concept, but using electrospraying was published in the same journal (Nanomaterials 2018, 8, 868; DOI:10.3390/nano8110868), and the authors also used b-carotene as a model molecule. The novelty of the present paper is low. Figure 2 is derived from Figure 1, so it would be better to combine it with Figure 1 as insets. Also, it is also highly suggested to put the SEM photos with the same magnification for better comparison. Please note that scale bars are not clear in the respective photos. All the experiments presented in the paper can be done in 2 days. I do not suggest the acceptance of the paper in the current form. I would suggest the following experiments to improve the quality of the paper before submission to any journal.

Since the main claim of the paper is the stabilization of compounds of high biological value, it would be better to use different sensitive food ingredients (4-5) and explore the effects of their incorporation on the fiber properties. Because of low b-carotene loading (1% w/w), TGA does not clearly show enhanced thermal stability. Thus, please increase the b -carotene loading (the authors can use other ingredients to enhance the loading capacity) and perform the test again. The in-depth study on their stabilization with encapsulation into microfibers at various concentrations is suggested. Figure 1 shows that the fibers are very brittle. The tensile properties of the fibers via DMA would be interesting to study. The dissolution/stability of the fibers in water should be studied.

Minor points

Line 95 “96° ethyl alcohol” should be replaced by “96% ethyl alcohol”. For viscosity measurements, what is the gap size between the plates and the viscosity is given at which shear rate? Line 115 “The injector based on a syringe pump with flow from 10 uL/h,” The flow speed is extremely low.. Is it should be 10 mL/h? There is a typo in Figure 2, y axis should be Frequency, not Frecuency

Author Response

M.S. Number: Nanomaterials-637349

Journal:  Nanomaterials

Corresponding Author:    RAGAZZO-SÁNCHEZ Juan Arturo.

Co-Authors:  Carla N. Cruz-Salas, Cristina Prieto, Montserrat Calderón-Santoyo, José M. Lagarón, Juan A. Ragazzo-Sánchez.

Title:Micro- and nanostructures of Agave fructans to stabilize compounds of high biological value via electrohydrodynamic processing.

__________________________________________________________________

Response to Reviewer 1 Comments

This article deals with the preparation of b-carotene encapsulated a high degree of polymerization agave fructans (HDPAF) microfibers by electrospinning. Although the concept of the paper, i.e., the stabilization of the b-carotene by embedding in microfibers might be interesting, the experimental results are very poorly presented. More importantly, a similar concept, but using electrospraying was published in the same journal (Nanomaterials 2018, 8, 868; DOI:10.3390/nano8110868), and the authors also used b-carotene as a model molecule. The novelty of the present paper is low.

R= The novelty of this paper lies in the production of fibers with this polysaccharide for the first time. Moreover, this material has shown protective properties, being interesting for the encapsulation of bioactive compounds. Therefore, this biopolymer could be an alternative to synthetic polymers in food, pharmaceutical and biomedical applications. On the other hand, β-carotene was selected as model bioactive compound, due to its high sensitivity and because of it is a really well-known compound, in this fashion comparisons with other matrices could be stablished. In addition, the previous work published in this research group enables us to stablish a comparison between the protective effect of the fibers and the capsules.

Figure 2 is derived from Figure 1, so it would be better to combine it with Figure 1 as insets. Also, it is also highly suggested to put the SEM photos with the same magnification for better comparison. Please note that scale bars are not clear in the respective photos.

R= As Referee 1 recommended, Figure 1 and Figure 2 have been combined, and SEM images with same magnification are presented (40 µm). In addition, images with higher magnification are shown to observe the surface of the fibers.

All the experiments presented in the paper can be done in 2 days. I do not suggest the acceptance of the paper in the current form. I would suggest the following experiments to improve the quality of the paper before submission to any journal. Since the main claim of the paper is the stabilization of compounds of high biological value, it would be better to use different sensitive food ingredients (4-5) and explore the effects of their incorporation on the fiber properties.

R= The aim of this paper is to show the potential of this material as a protective matrix and as an alternative to synthetic polymers in food, pharmaceutical or biomedical applications. β-carotene was selected as model bioactive compound, since it is a challenging bioactive due to its low physico-chemical stability. Nevertheless, this paper belong to a new research line in our group, in which other bioactive compounds extracted from plants with anti-inflammatory properties will be encapsulated in this matrix.

Because of low b-carotene loading (1% w/w), TGA does not clearly show enhanced thermal stability. Thus, please increase the b -carotene loading (the authors can use other ingredients to enhance the loading capacity) and perform the test again. The in-depth study on their stabilization with encapsulation into microfibers at various concentrations is suggested.

R= We totally agree with the reviewer that a loading of β-carotene of 1% is very low. However, regarding this molecule, the obtained loading capacity is a quite good value; taking into account that β-carotene presents low solubility in water, organic solvents and oils. Other authors have reported even lower loading capacities than the one reported in this paper. For example, Desobry et al. reported a loading capacity of 0.12% in the encapsulation of β-carotene into maltodextrin using spray drying as encapsulation technique (Journal of Food Science 62 (1997) 1158-1162), and Deng et al. reported a loading capacity of 0.1% in the encapsulation of β-carotene into soy protein isolate and OSA-modified starch by spray drying (Journal of Applied Polymer Science 131 (2014) 40399).

Regarding TGA results, the thermal stability of the fibers increased more than 50ºC. In terms of applications, this could mean that the product containing the particles could be subjected, for example, to a pasteurization process. 

Figure 1 shows that the fibers are very brittle. The tensile properties of the fibers via DMA would be interesting to study. The dissolution/stability of the fibers in water should be studied.

R= We totally agree with the reviewer, that the main disadvantage of these fibers is their high hygroscopicity and their brittleness (Lines 338-341). For this reason, we are performing fibers of Agave fructans in combination with other biopolymers in order to improve their hygroscopicity and their mechanical properties. This extensive work will be published soon.

Minor points

Specifics comments

Line 95 “96° ethyl alcohol” should be replaced by “96% ethyl alcohol”.

R=The sentence has been changed as suggested. Line 100.

For viscosity measurements, what is the gap size between the plates and the viscosity is given at which shear rate?

R=These values were added in response to the suggestion. Lines 111-114 and as a footnote in Table 1.

Line 115 “The injector based on a syringe pump with flow from 10 uL/h,” The flow speed is extremely low. Is it should be 10 mL/h?

R=The correction of the operating parameter values was performed. Lines 118-123.

There is a typo in Figure 2, y axis should be Frequency, not Frecuency

R= The word Frecuency has been changed by Frequency in the figure. Figure 2 was included as an inset in Figure 1.

Reviewer 2 Report

The manuscript entitled “Micro- and nanostructures of Agave fructans to stabilize compounds of high biological value via electrohydrodynamic processing” demonstrated that fructans could be a feasible alternative to replace synthetic polymers in the encapsulation of compounds of high biological value. The concept of the paper is fine; however, manuscript must be revised to improve its condition. The issues are listed as below.

Author may present biocompatibility of these nanofibres on human or animal cell lines or cells. This is important very important for biocompatibility test. Author must check viability of cells or cell lines for validation. Author must add statistical analysis in figure 2. Distribution of nanofiber diameters obtained……with B-carotene Quality of all figures are weak, author must improve quality of figures using professional software’s. I recommend author to revise all font size.

Author Response

M.S. Number: Nanomaterials-637349

Journal:     Nanomaterials

Corresponding Author:    RAGAZZO-SÁNCHEZ Juan Arturo.

Co-Authors:   Carla N. Cruz-Salas, Cristina Prieto, Montserrat Calderón-Santoyo, José M. Lagarón, Juan A. Ragazzo-Sánchez.

Title:Micro- and nanostructures of Agave fructans to stabilize compounds of high biological value via electrohydrodynamic processing.

__________________________________________________________________

Response to Reviewer 2 Comments

The manuscript entitled “Micro- and nanostructures of Agave fructans to stabilize compounds of high biological value via electrohydrodynamic processing” demonstrated that fructans could be a feasible alternative to replace synthetic polymers in the encapsulation of compounds of high biological value. The concept of the paper is fine; however, manuscript must be revised to improve its condition. The issues are listed as below.

Specifics comments

Author may present biocompatibility of these nanofibres on human or animal cell lines or cells. This is important very important for biocompatibility test. Author must check viability of cells or cell lines for validation.

R= The aim of this paper was to study the feasibility of high degree of polymerization Agave fructans (HDPAF) to produce micro-nanofibers through the electrospinning process, using a β-carotene as model compound, and study its protective properties. We totally agree with the referee that the biocompatibility test of this material into human or animal cell lines should be performed, and therefore, this will be the aim of our next paper. Moreover, the addition of different bioactive compounds, such as vegetal extracts with biological activity will be considered for further studies, comparing the permeability and anti-inflammatory properties before and after encapsulation. 

Author must add statistical analysis in figure 2.

R= Statistical analysis has been added. Figure 2 was included as an inset in Figure 1 as recommended by Reviewer 1.

Distribution of nanofiber diameters obtained……with B-carotene Quality of all figures are weak, author must improve quality of figures using professional software’s.

R= Quality of SEM micrographs has been improved and images showing the same magnification were used (40 µm), in order to ease the comparison. Additionally, images with a higher magnification were also shown.

I recommend author to revise all font size.

R= Font size has been checked in all document.

Reviewer 3 Report

The authors present an interesting study on the use of agave fructans for the incorporation of B-carotene in fibers. They demonstrate high loading efficiency, efficient fabrication, and enhanced thermo- and photostability. Some suggestions for inclusion are the following:

It is unclear from the Intro. what the final application is? Will these be used as a food/vitamin source in the current vehicle form?

Advantages of this process relative to others could be elaborated on.

Explicitly state the novelty and impact of this work. This comes across more in the Results/Discussion during comparisons, but is not as evident in Intro.

Fig. 1: make scale bar more apparent and b/d and c/e with the same scale.

Fig. 2: colors of squares in legends do not come across; also add error bars.

Thermograv. Analyses: how relevant are these high temperature stability experiments to the final application? During use, would conditions be within the range of 100-400 C?

Section 3.5: Please elaborate on: “This can be explained due to the nature of the compounds and their affinity between them”. What compounds, please be specific.

Line 288: describe electrospraying of capsules – assume you mean spraying the coating on capsules? How do you know the thickness of the capsule shell vs. distance of fiber surface to B-carotene?

Author Response

M.S. Number: Nanomaterials-637349

Journal:          Nanomaterials

Corresponding Author:    RAGAZZO-SÁNCHEZ Juan Arturo.

Co-Authors:      Carla N. Cruz-Salas, Cristina Prieto, Montserrat Calderón-Santoyo, José M. Lagarón, Juan A. Ragazzo-Sánchez.

Title:Micro- and nanostructures of Agave fructans to stabilize compounds of high biological value via electrohydrodynamic processing.

 _________________________________________________________________

Response to Reviewer 3 Comments

The authors present an interesting study on the use of agave fructans for the incorporation of B-carotene in fibers. They demonstrate high loading efficiency, efficient fabrication, and enhanced thermo- and photostability. Some suggestions for inclusion are the following:

Specifics comments

It is unclear from the Intro, what the final application is? Will these be used as a food/vitamin source in the current vehicle form?

R= Introduction was clarified as requested.  Lines 91-94.

Advantages of this process relative to others could be elaborated on.

R= The main advantages of the electrohydrodynamic processing with respect to conventional technologies is that this process can be carried out at room temperature, and the process does not require the use of toxic reactives, which could affect the stability of the bioactive compound or its subsequent application. Those advantages were already described in Lines 58-63.

Explicitly state the novelty and impact of this work. This comes across more in the Results/Discussion during comparisons, but is not as evident in Intro.

R= The novelty and impact of this work has been stated in the intro as requested by the referee. Lines 78-81.

Fig. 1: make scale bar more apparent and b/d and c/e with the same scale.

R= Two images are replaced on the same scale and their corresponding magnifications

Fig. 2: colors of squares in legends do not come across; also add error bars.

R= The color has been corrected and the error bars were added. Figure 2 was included as an inset in Figure 1, as requested by another referee.

Thermograv. Analyses: how relevant are these high temperature stability experiments to the final application? During use, would conditions be within the range of 100-400 ºC?

R= The thermogravimetric analysis was performed to characterize the thermal protection of fructans as encapsulating material. We have used the parameters of a conventional test, heating ramp of 25 to 600 °C at a speed of 5 °C/min. This analysis is interesting in order to know if the capsules will resist a post-processing step when included on a product, such as pasteurization or any other process that requires temperature.

Section 3.5: Please elaborate on: “This can be explained due to the nature of the compounds and their affinity between them”. What compounds, please be specific.

R= Specification has been made in Line 286.

Line 288: describe electrospraying of capsules – assume you mean spraying the coating on capsules? How do you know the thickness of the capsule shell vs. distance of fiber surface to B-carotene?

R= The experimental results show twice the oxidation in the fibers than in the capsules, although the b-carotene content is 10 times higher in the fibers. Therefore, the biopolymer:bioactive ratio is lower in the fibers, and we can expect that the thickness and the protective effect to be reduced. The photostability of b-carotene in the fibers could also be affected by the fact that the fibers have a greater surface area exposed to light. This explanation was clarified in the manuscript. Lines 304-310.

Reviewer 4 Report

In the Fig.2 the distributions are presented in different collors, but in the title of figure both are presented in the same collor.

To describe the manufactured fibres as nanofibres is not correct. All fibres have much higher diameter as 100 nm or even 500 nm. So, in this work not nanofibres are developed. I suggest to change in all text the term of nanofibres in the term micro- nanofibres, as is used in the title of paper.

Author Response

M.S. Number: Nanomaterials-637349

Journal:        Nanomaterials

Corresponding Author:    RAGAZZO-SÁNCHEZ Juan Arturo.

Co-Authors:     Carla N. Cruz-Salas, Cristina Prieto, Montserrat Calderón-Santoyo, José M. Lagarón, Juan A. Ragazzo-Sánchez.

Title:Micro- and nanostructures of Agave fructans to stabilize compounds of high biological value via electrohydrodynamic processing.

__________________________________________________________________

Response to Reviewer 4 Comments

In the Fig.2 the distributions are presented in different colors, but in the title of figure both are presented in the same color.

R= The color has been corrected.

To describe the manufactured fibres as nanofibres is not correct. All fibres have much higher diameter as 100 nm or even 500 nm. So, in this work not nanofibres are developed. I suggest to change in all text the term of nanofibres in the term micro- nanofibres, as is used in the title of paper.

R = The term nanofibers has been corrected as suggested by the reviewer throughout the text.

Round 2

Reviewer 1 Report

The manuscript can be accepted in the present form.

Reviewer 2 Report

I recommend to accept this manuscripts author revised it.